# Effects of summer schools: Who benefits the most?

**Mélanie Monfrance** [1]*, **Carla Haelermans**[1], **Trudie Schils**[2]

**1** Research Centre for Education and the Labour Market (ROA), School of Business and Economics, Maastricht University, Maastricht, The Netherlands, **2** Department of Macro, International and Labour Economics (MILE), School of Business and Economics, Maastricht University, Maastricht, The Netherlands

* melanie.monfrance@maastrichtuniversity.nl

## Abstract

This study investigates whether publicly funded summer school programs in secondary education are of substantive meaning for the math performance of students from lower socioeconomic backgrounds. More specifically, we explore whether this is the case when the summer programs are not explicitly aimed at low-SES students. In this context, we investigate whether summer schools in the Netherlands can decrease inequalities of opportunities in education. We use administrative data from Dutch secondary schools. To analyse the effect of the intervention we apply a Difference-in-Difference analysis in combination with matching estimation techniques. The results indicate that there is an overall modest effect of participation in a summer school. When analysing the SES groups separately, we observe a positive effect of participation for all three SES groups. However, the effect seems less strong for participants in the lowest and middle SES group compared to the highest SES group.

## Introduction

Despite extensive research and policy focus on social inequalities in education and inequalities in educational opportunities, they remain a highly relevant societal problem [1, 2]. A vast amount of literature has documented that a lower socioeconomic background is associated with lower levels of school achievement [3–5] and increasing disparities herein throughout students' lives [6]. Literature also shows that due to their socioeconomic background, children with comparable levels of cognitive potential do not reach the same level of education when they graduate [7, 8]. These inequalities in educational opportunities are often explained by differential access to economic, cultural, and social resources [9].

One crucial period when disparities according to socioeconomic background become even more pronounced is during the summer vacation when school resources are unavailable, placing greater dependence on family resources. Children from higher socioeconomic backgrounds tend to have more resources available, providing them with more enrichment and learning activities during the summer [10, 11]. Consequently, a major focus of previous research has been on the inequalities in learning that increase during the summer vacation period. This growing gap during the summer holiday period for students with a lower

**Funding:** The data collection for this study was funded by the VO-raad (secondary education, sector organisation in the Netherlands). The funders had no role in study design, data collection and analysis, decision to publish, or preparation of the manuscript.

**Competing interests:** The authors have declared that no competing interests exist.

socioeconomic background is often referred to as 'summer setback,' 'summer slide' or 'summer learning loss' [12–15]. An elaborate body of literature on summer learning in the United States suggests that inequalities increase during the summer holiday period [11, 13, 14] specifically for math [13, 15, 16]. More recently, however, there has been some debate about the magnitude of summer learning loss and resulting inequalities between students [17–19]. In European literature addressing summer learning loss, a decline in performance was observed for skills in math and spelling, and stagnation or even gain was observed for reading [20–22].

As a response to these disparities, over the past decades, policy initiatives were taken to provide free additional education opportunities for children from lower socioeconomic backgrounds. In the United States, for example, various publicly funded academic school-based programmes that target socioeconomically disadvantaged students are organised during the summer [23]. These so-called summer school programmes have been introduced to avoid grade repetition and improve the academic outcomes of disadvantaged students [24–26]. Generally, these programmes focus on reading and math because they are considered crucial domains for student learning [27].

Numerous studies have been conducted to evaluate the effects of school-based summer school programmes on math, literacy, or both [24, 26, 28–31]. Following from this vast amount of studies, several systematic reviews, that include the aforementioned and other studies, have addressed the impact of summer schools. The first meta-analysis was performed by [32] Cooper et al. [32] in 2000 and found positive effects of summer school interventions on math and literacy performance. Although Cooper et al. [32] argue that summer schools positively impact all participants, middle-class students show larger positive effects than students from lower socioeconomic backgrounds. Following Cooper et al.'s meta-analysis, Kim and Quinn [33] found positive results of summer school participation on student performance in math. In contrast to Cooper et al. [32] they compare students from lower and higher socioeconomic backgrounds and find that the positive effects specifically hold for students from lower socioeconomic backgrounds. Their explanation for this differential finding is that the summer school treatment/control contrast for middle-income students may be weaker now than in the past due to the higher investment by wealthier parents today [33]. A more recent meta-analysis by Xie et al. [34] reports positive mean effects for reading and math. However, these results are not confirmed in their meta-regression (which could be a statistical artifact since they only include 19 studies). The most recent meta-analysis by Lynch et al. [35], which includes the most contemporary experimental and quasi-experimental studies of summer programmes in mathematics, find that summer programmes positively affect mathematics achievement outcomes. These effects were comparable for students from lower and higher socioeconomic backgrounds. Furthermore, the effect of summer school programmes appears to depend on their time intensity and characteristics. The programmes tested typically lasted several weeks (ranging between three and six weeks) and were effective only if students participated for the entire period [8, 24, 32]. In addition, Merry et al. [36], argue that it is essential to have high-quality summer school education for students from lower socioeconomic backgrounds to benefit. For example, McCombs et al. [8] found that literacy, instructional quality, teacher grade-level experience, and site orderliness were associated with better outcomes.

Summer school programmes, often explicitly focussing on disadvantaged students, are common practices in the United States. In the Netherlands, however, there is no longstanding tradition of (publicly funded) summer school programmes except in higher education. Based on the summer school policies and related positive findings in the United States, policymakers took the initiative in 2015 to subsidize the organization of summer schools in secondary education as they were concerned about the high rate of grade retention in the Netherlands. These summer school programmes in the Netherlands did not specifically target disadvantaged

students. By examining whether these summer school programmes affect performance and whether this effect differs by socioeconomic background, this study contributes to the existing research on summer schools. From a policy perspective, this is of importance since this provides insights into possible unintended policy effects, for example, when students from lower socioeconomic backgrounds do not benefit or benefit less from a policy compared to students from higher socioeconomic backgrounds. Furthermore, the concerns about increasing inequalities of opportunities in education make this study particularly relevant. Given the greater impact of summer schools on math performance [16, 37] and the prevalence of math as a subject in Dutch summer schools [38, 39], this study focuses on math performance and is structured around two main research questions:

Question 1: What is the effect of participation in a summer school on the average performance in math?

Question 2: Does this effect differ between students with lower socioeconomic backgrounds and students with a higher socioeconomic background?

## Summer schools in the Netherlands

Summer schools in the Netherlands were funded by the National government. Schools were eligible to apply for financial support in both the 2015–2016 and the 2016–2017 school year. The funding allocation was based on the number of participating students, with a fixed funding amount per participating student, where each school received support proportional to its budgeted student count. This funding model ensures that differences in financial capacity did not hinder schools from organizing summer school programmes.

Furthermore, the Dutch summer schools had specific characteristics, they were intensive remedial teaching interventions targeting only students at the margin of repeating a grade. Students who have a vast amount of deficits and are to repeat the grade anyway, are not part of the summer school target population. The summer school intervention took place at the start of the summer break. Attendance in a summer school was non-compulsory, and coaching and personalized practicing are offered regarding aspects that the student's teacher identified as beneficial. This tries to ensure that the most critical deficiencies are dealt with. Besides a couple of basic rules, that is, The Summer schools needed to at least partly take place during the regular holiday period and schools were not allowed to request any charge for participation, schools are free to organize these remedial programmes. Therefore, a detailed overview of each program's setup is unavailable. On average, these programmes last about seven days (maximum of two weeks) of six hours and target one or two subjects for most students. Furthermore, mostly non-qualified teachers (that is students still in teacher education) were teaching in the summer schools. How summer schools in the Netherlands are organized cannot be separated from the institutional context of Dutch education. The focus on preventing grade retention is very much related to the Dutch system, where students have to repeat a whole school year if they have a poor performance and specific deficits in (only) three courses.

Comparable to the publicly funded summer schools in the United States, there is no charge for participation in the Dutch summer schools. This implies that summer schools are accessible for all students from all grades. However, contrary to the summer schools targeting disadvantaged students in the United States, summer schools in the Netherlands do not specifically target disadvantaged students.

## The Dutch education system

In contrast to countries without external differentiation, such as the United States or Sweden, but comparable to other European countries such as Germany and Austria, one of the main

characteristics of the Dutch education system is external differentiation, also referred to as tracking. Pupils attend primary education between age 4 and 12 and secondary education until a secondary degree is obtained (age 12–18, depending on the track followed). Based on a standardized test and teacher advice, a track recommendation is given at the end of primary school before starting secondary education in different tracks. Secondary education begins in seventh grade when students are around 12 years old. Within regular secondary education, three different levels of education are offered: pre-vocational secondary education (within this, a distinction is made between two more theoretical and two more practically oriented tracks), general secondary education preparing for universities of applied sciences, and pre-university education (Known by the Dutch acronyms *vmbo*, *havo* and *vwo*). About half of Dutch students follow the pre-vocational track, about a quarter of the students follow the general secondary track, and about one-fifth the pre-university track. Depending on the chosen track, Dutch secondary education has a variable length. The pre-vocational track lasts four years, the general track five years, and the pre-academic track six years. Students are obliged to meet the requirements set by the school of the track for all subjects; otherwise, they have to repeat the grade. All students in the general and pre-university tracks take the subjects Dutch, English, and math. For the pre-vocational track, taking math depends on the chosen program in the third and fourth year (age 15–16).

## Data, sample selection and variables

### Data

The data used in this research were obtained from the administrative systems of Dutch secondary schools that organized a summer school during either the 2015–2016, 2016–2017, or both school years. Schools were asked to provide student information from their administrative records for all students of the grade levels in which participating students were located. This ensured that we not only have information on participants but also on students in the same track and grade level in the year in which the summer school was organized. Information was extracted regarding personal characteristics (e.g., gender, age, educational track, and year) and study progress (grade retention, grades for the periodic student reports before participation in the summer school, and the first-grade overview of the following school year). The data of both subsequent years were merged to ensure more power. For schools that organized a summer school in both years, this implies that some students are observed twice in the dataset. In the course of the analyses, we explicitly deal with this merge by removing duplicates at the student level and including a control variable for the year of observation.

Participants and parents were informed about the purpose of the data collection and were given the option to decline participation. For the data collection and data management, a Privacy Impact Assessment was conducted, and privacy measures were taken accordingly. The Graduate School of Business and Economics at Maastricht University granted an exemption from the Institutional Review Board (IRB) for the data collection.

### Analytic sample

The merged dataset comprised 37,789 students and 1,501 summer school participants from 66 schools. Based on this merged dataset, we selected our analytic sample in different stages. First, we selected students for whom we have complete information about study progress, that is, grades for the periodic student report before the start of the summer school, grade repetition, and the first-grade overview of the following school year.

Second, we excluded students for whom we missed age and socioeconomic background information. Third, we excluded students who had an average of 7.5 or higher from the sample

to have more comparability between the group of participants and the control group of non-participants in summer schools since no students with such a high average grade actually participated and therefore could not find similar counterparts in the other group anyway. Next, we excluded students who appeared twice in the data while not being completely identical (e.g., on grades); for non-participants, we randomly determined which observation to keep, whereas, for participants, we only kept the observation of the student participating in the summer school.

Finally, only students who participated in math were included as summer school participants. Students who participated in summer schools for other courses were excluded. By eliminating these students, we ensured that no summer school participants were included in the group of non-participants. Tests showed that the selected sample is not likely to be biased on observable characteristics. To ensure that students who participated in summer schools for math are not a selective group compared to the other participants we checked with independent t-tests if the participants for math and other subjects were comparable on observed characteristics (where we transformed categorical variables into dummy variables). Furthermore, we looked at standardized mean differences for meaningful differences with respect to magnitude. The results of these tests indicate that there are no significant/meaningful magnitude differences between the participants for the control variables gender, education track, track advice, age, SES group, and grade level.

Our final analytic sample comprised 15,819 students from 55 schools, of which 285 students participated effectively in summer school. This sample is representative of the national sample [40] in terms of the average student performance (sample: M = 6.6, SD = .13; national average: M = 6.6, SD = .16) and percentage of grade retention (sample: M = 5.8%, SD = .02; national average: M = 7.1%, SD = .10). Regarding average school size (sample: M = 1211.1, SD = 495.92; national average: 677.6, SD = 513.5) and related to this average number of teachers employed in full-time equivalents (sample: M = 213.4, SD = 116.7; national average: M = 124.3, SD = 102.1) our analytic sample is not entirely representative. The sample consists, on average, of larger schools; therefore, some precautions must be taken into account about external validity since the results are to some extent less representative for smaller schools.

## Variables

**Math GPA.** We used the math GPA (Grade Point Average) in the period prior to the summer school and the math GPA of the first period in the following school year, i.e., after the summer school took place. The math scores were calculated based on the grades for all the written math tests during that period. The outcome variable of interest is the math GPA in the first period of the school year after the summer school took place. Within a school, teachers in the same department develop the tests together and discuss the overall grading. The tests within a school are therefore the same for all the students, which (partly) eliminates a teacher effect for the given tests.

In the Netherlands, a ten-point grading system is used. In this system, 1 is the lowest grade and 10 is the highest. On final lists, grades are usually rounded off to half points. The passing grade is 5.5 (rounded off to 6). An overview of how Dutch grades are assigned and comparison with American grades can be found in the appendix (see S1 Table).

**SES.** Because of privacy regulations, no direct indicator of students' SES such as parental education level was available in the obtained administrative data. Therefore, students' SES is measured by using an indicator for the status score of the four-digit postal code area where a student lives (For an example of how a SES indicator is included at the postal level, see [41]). Status scores are calculated by the Netherlands Institute for Social Research (SCP) and indicate

the social status of a neighbourhood compared to other neighbourhoods in the Netherlands. The social status of a neighbourhood is derived from a number of characteristics of people who live in the neighbourhood: education level, income, and labour market position. The status scores are available for 2016 and 2017 [42] and linked to the two cohorts (school year 2015–2016 and 2016–2017) of participants. Subsequently, three consecutive categories for SES are distinguished based on three quantiles. Technically this means that we are analysing students from low-SES to high-SES communities, although for reasons of brevity we refer to them as low-SES to high-SES students. Furthermore, different classifications of the SES-categories are tested as a robustness check. These include, mean plus/minus one standard deviation, mean plus/minus half of a standard deviation and four equal groups. The results are in accordance with the results presented in this paper and can be found in the appendix (see S2–S4 Tables).

**Control variables.** Several student characteristics are included as control variables. *Gender* (female = 1, male = 0), *education track* (pre-vocational = 1, general = 2, pre-academic = 3, other, that is combination track = 4, unknown = 5), *track advice (*pre-vocational practical track = 1, pre-vocational theoretical track = 2, general track = 3, pre-academic = 4, unknown = 5), *grade repetition* (grade repetition subsequent school year = 1, no retention = 0), *year of participation* (2017 = 1, 2016 = 0), *age* in months and *grade level*.

The descriptive statistics for the variables included in the analysis are shown in Table 1. In Table 1 we observe that most of the students included are either in their third or fourth year of secondary education (i.e. grade 9 or 10, age 15–16) and in the general education track. This is also observable for the track advice. About 2 percent of the students included in the data have participated in the summer school and in total almost 8 percent of all students repeated a grade. When comparing the descriptive statistics between summer school participants and non-participants, based on an independent t-test, we see that there is a significantly higher percentage of males and students with a low SES participating in the summer schools. Furthermore, as can be expected, the percentage of grade repetition is also significantly higher among the participant group. For the other characteristics no major differences are observed.

## Methods and results

We start our analysis with descriptive statistics providing a first indication of the relationship between summer school participation and math GPA. Next, we use a Difference-in-Differences (DiD) estimation technique that compares the average growth in grades for mathematics between summer school participants and non-participants. Finally, we perform a DiD-analysis in combination with matching as a robustness check. We perform the analyses separately for the distinguished SES group to test for differential effects for SES. We standardize the outcome variable for interpretation by using z-scores (mean = 0 standard deviation = 1). Further elaboration on the used methods of analysis is included in each respective section.

### Descriptive results

Strong and robust patterns can be visible in descriptive graphs and tables [43]. Therefore, we start the analysis by presenting simple descriptive statistics where we compare the math GPA for participants in summer schools with the math GPA for non-participants, both before participation and in the first term of the new school year. Presenting descriptive tables for math GPA distinguishing between participants and non-participants gives a first indication of the possible effect of summer school participation. If the intervention is effective, we expect to observe a more pronounced change in the math GPA before and after participation in the summer school.

**Table 1. Descriptive statistics for total population and participants and non-participants separately.**

| | | Percentage | | |
|---|---|---|---|---|
| | N Total | Total | Participants | Non-participants |
| **Sex** | | | | |
| Female | 7817 | 49.4 | 36.1 | 46.5 |
| Male | 7326 | 46.3 | 59.3 | 49.2 |
| Unknown | 676 | 4.3 | 4.6 | 4.3 |
| **Year of secondary education** | | | | |
| 1 | 1046 | 6.6 | 4.9 | 6.6 |
| 2 | 3762 | 23.8 | 25.3 | 23.8 |
| 3 | 4913 | 31.1 | 32.3 | 31.0 |
| 4 | 4722 | 29.9 | 31.9 | 29.8 |
| 5 | 1376 | 8.7 | 5.6 | 8.8 |
| **Education track** | | | | |
| Pre-academic | 4526 | 28.6 | 24.6 | 28.7 |
| General | 6985 | 44.2 | 46.7 | 44.1 |
| Pre-vocational | 3759 | 23.8 | 25.6 | 23.7 |
| Other | 485 | 3.1 | 3.2 | 3.1 |
| Unknown | 64 | 0.4 | 0.0 | 0.4 |
| **Track advice** | | | | |
| Pre-academic | 3443 | 21.8 | 21.4 | 21.8 |
| General | 7577 | 47.9 | 47.7 | 47.9 |
| Pre-vocational theoretical track | 2578 | 16.3 | 15.4 | 16.3 |
| Pre-vocational practical track | 985 | 6.2 | 7.4 | 6.2 |
| Unknown | 1236 | 7.8 | 8.1 | 7.8 |
| **SES group** | | | | |
| SES-1 | 4843 | 30.6 | 38.3 | 30.5 |
| SES-2 | 5201 | 32.9 | 32.6 | 32.9 |
| SES-3 | 5775 | 36.5 | 29.1 | 36.6 |
| **Grade repetition** | | | | |
| No | 14577 | 92.1 | 87.0 | 92.2 |
| Yes | 1242 | 7.9 | 13.0 | 7.8 |
| **Summer school participation** | | | | |
| Yes | 285 | 1.8 | | |
| No | 15534 | 98.2 | | |
| **Year of summer school** | | | | |
| 2016 | 8837 | 55.9 | | |
| 2017 | 6982 | 44.1 | | |
| *N* | 15,819 | | | |

In Table 2, we observe that participants in summer schools are, on average lower performing students. This lower performance of participants is in line with our expectations, given that this lower performance ensures that these students are potential summer school

**Table 2. Descriptive results Math GPA for summer school participants and non-participants.**

| | Participants | Non-participants |
|---|---|---|
| Z Math GPA before summer school | -1.14 | 0.02 |
| Z Math GPA first term | -0.85 | 0.02 |
| Difference before and after summer school | 0.29 | 0 |

**Table 3. Descriptive results Math GPA for summer school participants and non-participants by SES group.**

|  | Participants | | | Non-participants | | |
|---|---|---|---|---|---|---|
|  | SES1 | SES2 | SES3 | SES1 | SES2 | SES3 |
| Z Math GPA before summer school | -1.21 | -1.09 | -1.12 | -0.02 | 0.03 | 0.05 |
| Z Math GPA first term | -0.95 | -0.87 | -0.68 | -0.03 | 0.05 | 0.02 |
| Difference before and after summer school | -0.26 | -0.22 | -0.44 | 0.01 | -0.02 | -0.02 |

candidates and the ones who potentially gain the most from participation in summer schools. In Table 2, we also observe that the math GPA of participants in the first term is, on average three-tenths of a standard deviation higher than in the period before the summer school, while for non-participants, the math GPA before the summer school and after the summer school seems to remain unchanged. This indicates that summer school participants experience a (stronger) growth in their math performance after participating in the summer school.

When we distinguish between different SES groups in Table 3, we observe that for non-participants, there is almost no difference between math GPA before and after the summer school. For participants we see that the math GPA on average is also higher ranging between 0.2 and 0.4 standard deviation, and that the average is higher for the highest SES group.

However, simply comparing the change in the GPA before and after the summer school programme between participants and non-participants can be problematic since possible selection bias in participation is not taken into account. In the following sections, we analyse the effect of the intervention while simultaneously taking into account possible selection bias.

## Difference in Difference analysis

Ideally, we would randomly assign students to the group of participants and non-participants to achieve comparability between these two groups to estimate the effect of participation in summer schools. However, participation in summer schools was not random, as participation was voluntary. Although all margin students in organizing schools were invited, not all accepted the invitation. Note that students could only opt-out and not opt in, so non-invited students could not participate. To account for selection effects, we use a Difference-in-Difference estimation technique. The DiD-method compares the change in outcomes over time between the group of participants (treatment group) and non-participants (comparison group) [44]. Herewith, we aim to investigate whether the development in performance is more favorable for participants than non-participants. The DiD-approach allows us to remove bias in the post-intervention comparisons between participants and non-participants that could result from permanent differences between these groups [44]. The DiD-model (Table 4 and Fig 1) compares the change in math performance over time (before and after the intervention) of summer school participants with the change in math performance of non-participants. First, the difference in performance before the summer school (A in Table 4) and after the summer school (B in Table 4) is calculated for participants (B-A in Table 4). Next, the difference between the performance before the summer school (C in Table 4) and after the summer

**Table 4. The Difference-in-Difference method.**

|  | Before the summer school (time = 0) | After the summer school (time = 1) | Difference |
|---|---|---|---|
| Participants (treated = 1) | A | B | B-A |
| Non-participants (treated = 0) | C | D | D-C |
|  |  |  | DiD = (B-A)–(D-C) |

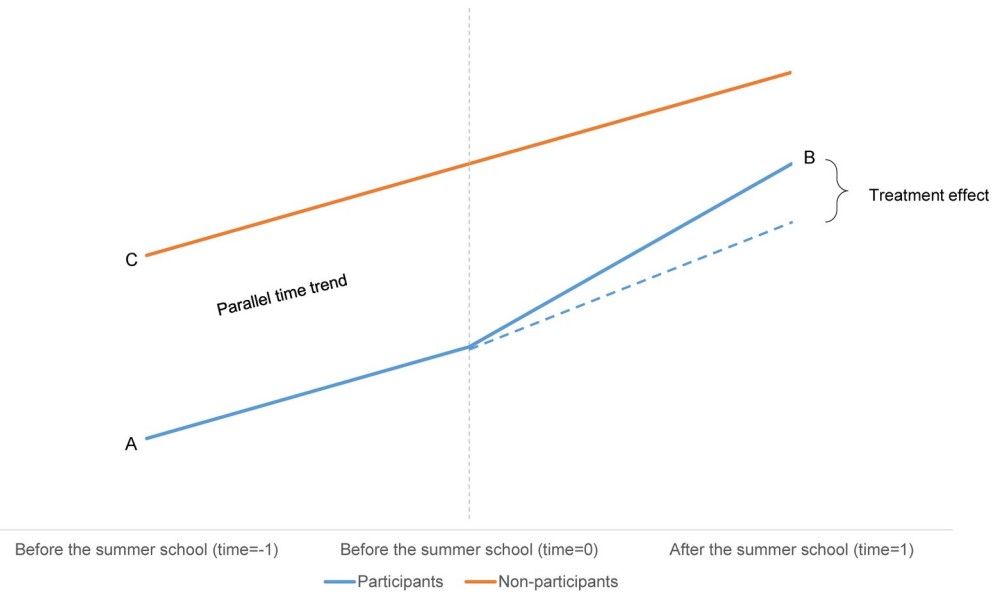

**Fig 1. Difference-in-Differences: A graphical explanation.**

school (D in Table 4) is calculated for non-participants (D-C in Table 4). Based hereupon, the difference between the difference in performance for the participant group (B-A) and the difference in the performance of the non-participant group (D-C) are compared, which provides the estimate of the DID-model (DiD in Table 4).

To estimate the DiD-estimator, we perform an Ordinary Least Squares (OLS)-regression including a dummy indicator for time (before participation = 0, after participation = 1), a dummy for summer school participation (participants = 1, non-participants = 0), and an interaction term between the time and participation dummy. This DiD-estimator is the primary variable of interest and indicates whether or not the average growth over time between the two groups is significantly different. Subsequently, we perform the analysis separately for SES groups to analyse if the effect of participation differs across SES groups. We included clustered standard errors at the school level since observations within a school are not independent of each other. Literature, for example, shows that teacher effectiveness is systematically different between schools [45, 46].

Before estimating the DiD-estimator, the parallel time trend assumption must be checked, which is an essential assumption of the DiD-model [47]. This assumption ensures the internal validity of the DiD-model and refers to the difference between the participants and non-participants being constant over time prior to the treatment (e.g., having a parallel time trend, see Fig 1). This parallel time trend is crucial to this method as the DiD can eliminate constant and constant differences between treatment and control groups but cannot eliminate differences between the treatment and control groups that change over time. This is important because the DiD assumes that in the case of the absence of the summer school program, the differences between post- and pre-test between participants and non-participants would have remained the same.

Fig 2 visually indicates the nine months before selection for the summer schools and ranges from +3 and -3 standard deviations from the mean, referring to a full normal distribution. The trends in math GPA for each grading period are about equal for students who participate in summer school and those who do not. As such, we can conclude that the parallel time trend

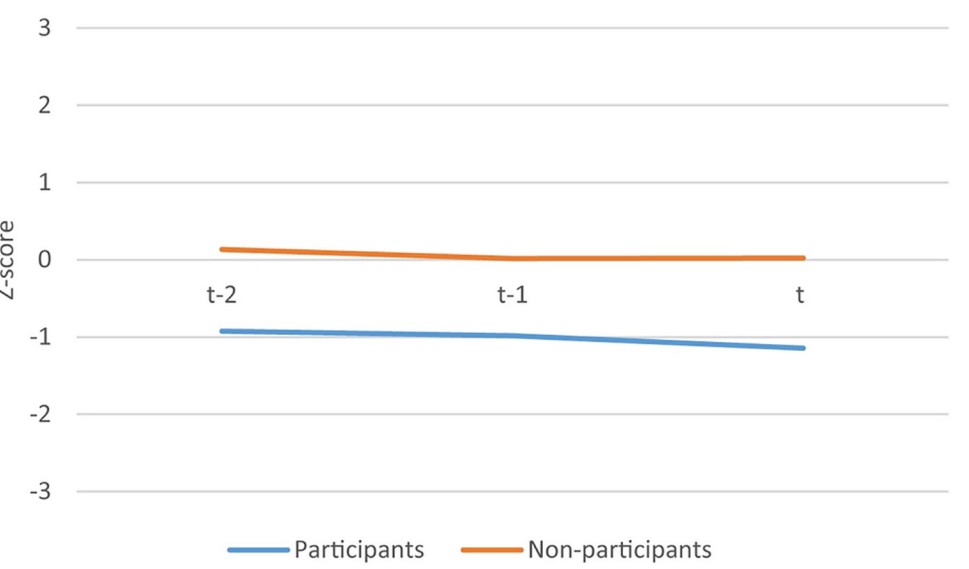

**Fig 2. Z-score Math GPA during grade periods.**

assumption is not violated. The parallel time trend assumption is also checked for SES group separately. This assumption is also not violated for the different SES groups. for reasons of brevity we do not present these figures separately.

Another important assumption related to the DiD-analyses is the Stable Unit Treatment Value (SUTVA) which relates to the potential spill-over effects of the intervention. The SUTVA has two components; the first one refers to interference, the treatment in one group should not affect the outcomes in the non-treated group. The second relates to treatment versions; for the treatment and the control group, there should only be one form of treatment. Both components of the SUTVA assumption will likely not be violated in the summer school intervention. About interference, the summer schools took place at the beginning of the summer break when the summer break started for non-participants. After the summer school, the summer break lasted at least 4 weeks for the participants, making spillover effects less likely, as they only saw non-participants again after the break at the start of the school year. Furthermore, the summer school intervention focused on individual deficits from the past school year, making it less relevant for non-participants. Regarding the second component, there were only two versions of the treatment, whether you participated in a summer school or not. In the hypothetical (and unlikely) case that there would still be some spill e.g. because participants and non-participants spend time together or do homework together in the subsequent school year, this would only result in an underestimation of the effect, making the differences that we find in this study between both groups of students smaller than they are in reality.

Table 5 presents the results of the DiD-estimates for the entire sample including clustered standard errors at the school level. The first model includes the DiD-estimator (interaction between time and summer school participation) without control variables and indicates that the average growth in math GPA between the grade period before and after the summer school is stronger for participants in summer schools compared to non-participants. The effect size is .304 ($p<0.01$) of a standard deviation difference in growth for participants compared to non-participants. The second model including the control variables confirms this result, with respect to both magnitude and significance of the coefficient.

Table 6 presents the results of the DiD-estimates for SES groups separately. The results for the separate SES groups show that the DiD-estimator is significant for all three SES groups,

**Table 5. OLS regression of the DiD-estimates for Z Math GPA.**

|  | Model 1 | Model 2 |
|---|---|---|
| Constant | 0.044 | 1.241*** |
|  | (0.030) | (0.202) |
| Summer school | -1.155*** | -1.095*** |
|  | (0.044) | (0.049) |
| Time indicator | -0.005 | -0.005 |
|  | (0.019) | (0.019) |
| Summer school * Time indicator | 0.304*** | 0.304*** |
|  | (0.082) | (0.082) |
| Control variables |  | yes |
| Observations | 31,638 | 31,638 |
| Number of clusters | 55 | 55 |
| R-squared | 0.019 | 0.065 |

* $p < 0.10$

** $p < 0.05$

*** $p < 0.01$. Standard errors in parentheses are clustered at the school level

Note, included control variables are: gender, age, SES group, grade level, education track, track advice, grade repetition, and year of participation

albeit the significance level differs. For the lowest SES group, this indicates a stronger average growth of 0.274 (P<0.10) of a standard deviation for participants in this SES group compared to non-participants. The stronger average growth is comparable for the middle SES group with 0.192 (P<0.10) of a standard deviation. For the highest SES group, the observed effect is stronger, with a standard deviation of 0.463 (P<0.01). The results of the robustness check, including different distinctions for SES groups, confirm these findings. For the lowest SES group, it seems that it is specifically the upper part within this group where significant effects are

**Table 6. OLS regression of the DiD-estimates for Z Math GPA separately for SES groups.**

|  | SES-1 | | SES-2 | | SES-3 | |
|---|---|---|---|---|---|---|
|  | Model 1 | Model 2 | Model 1 | Model 2 | Model 1 | Model 2 |
| Constant | 0.007 | 0.718*** | 0.051 | 1.775*** | 0.086** | 1.148*** |
|  | (0.039) | (0.244) | (0.031) | (0.304) | (0.043) | (0.346) |
| Summer school | -1.170*** | -1.095*** | -1.103*** | -1.086*** | -1.170*** | -1.092*** |
|  | (0.083) | (0.075) | (0.077) | (0.087) | (0.075) | (0.099) |
| Time indicator | -0.006 | -0.006 | 0.018 | 0.018 | -0.026 | -0.026 |
|  | (0.025) | (0.025) | (0.027) | (0.027) | (0.024) | (0.024) |
| Summer school * Time indicator | 0.274* | 0.274* | 0.192* | 0.192* | 0.463*** | 0.463*** |
|  | (0.152) | (0.152) | (0.107) | (0.107) | (0.091) | (0.091) |
| Control variables |  | yes |  | yes |  | yes |
| Observations | 9,686 | 9,686 | 10,402 | 10,402 | 11,550 | 11,550 |
| Number of clusters | 51 | 51 | 55 | 55 | 51 | 51 |
| R-squared | 0.024 | 0.059 | 0.019 | 0.074 | 0.014 | 0.075 |

* $p < 0.10$

** $p < 0.05$

*** $p < 0.01$. Standard errors in parentheses are clustered at the school level

Note, included control variables are: gender, age, grade level, education track, track advice, grade repetition, and year of participation

observed. The results of these robustness checks for SES group can be found in the appendix (see S2–S4 Tables).

## Robustness check: Matching and Difference-in-Difference analysis

As mentioned in the previous section, summer school participation is likely not random. The DiD-method tries to compensate for this by assuming that when the parallel trend holds, this will eliminate the (non-observed) differences between the participant and non-participants. While the parallel time trend holds in this study, the period over which the trend is observed is relatively short, with the risk of showing a snapshot biased by unobserved characteristics. To control as much as possible for potential selection bias due to differences in non-observable characteristics, we use a propensity score matching technique in combination with the previously used DiD-method as a robustness check. Combining the DiD-technique with the Matching approach reduces this possible confounding bias by selecting and comparing participants and non-participants that are comparable in observable characteristics [48].

We match on observed pre-treatment characteristics (the control variables on which we match students are: gender, age, SES group, grade level, track advice, and additional characteristics: school, average grade for all courses, and the number of shortage points prior to the selection moment). The number of shortage points are calculated based on the distance between the grade and the passing grade. Between 4.5 and 5.4 counts as one shortage point, between 3.5 and 4.4 counts as two shortage points and under 3.5 counts as three shortage points. The students were randomly ordered and matched using the nearest neighbour (NN) matching algorithm with replacement and oversampling of 1 to 10. For the analyses distinguishing between the different SES groups the matching procedure was applied separately within each respective SES group. Each participant is matched to 10 non-participants that resemble the student based on the pre-treatment characteristics, and untreated students can be used more than once as a match. The NN 1 to 10 matching technique allows for a larger control group and therefore provides more power than, for example 1 to 1 matching. Furthermore, the NN technique, compared to the Kernel Matching (KM) or Local Linear Matching (LLM) techniques, ensures that for the control group, selected students are closest to the treated students in terms of propensity scores, while the KM and LLM techniques use weighted averages of (nearly) all students [49]. Both the KM and LLM techniques would, in this case, entail that not all used observations are good matches, especially given that there are always high-performing and low-performing students. Given that the data contains all students of the same grade level as the participating students, it is likely that we find sufficient comparable students that did not participate using the NN matching technique. These matched students comprise a more comparable control group than when we construct a control group that includes *all* non-participants. The success of the matching procedure is tested by the check of balance between the treatment and control groups. The results of the t-tests, after transforming the categorical variables into separate dummy variables, indicate that there are no significant differences between the treatment (participants) and control (non-participants) groups for the pre-treatment background characteristics (gender, education track, track advice, age, SES group, and grade level). A significant difference is, however, observed between the treatment and control groups for the characteristics that represent treatment group candidates, namely, the number of shortage points and GPA for all courses. However, a joint F-test shows that these variables jointly cannot significantly predict participation in a summer school which provides confidence about the matching. Furthermore, the differences between treatment and control groups are confirmed when applying the Bonferroni correction.

**Table 7. OLS regression of the DiD-estimates for Z Math GPA on the matched sample.**

|  | Model 1 | Model 2 |
|---|---|---|
| Constant | 0.057 | 0.535 |
|  | (0.048) | (0.388) |
| Summer school | -0.705*** | -0.754*** |
|  | (0.053) | (-0.051) |
| Time indicator | -0.026 | -0.026 |
|  | (0.043) | (-0.043) |
| Summer school * Time indicator | 0.206*** | 0.206*** |
|  | (0.062) | (-0.062) |
| Control variables |  | yes |
| Observations | 4,612 | 4,612 |
| Number of clusters | 43 | 43 |
| R-squared | 0.040 | 0.121 |

Note: included control variables are: gender, age, SES group, grade level, education track, track advice, grade repetition, and year of participation

The success of the matching procedure, tested with a t-test, joint F-test and Bonferroni correction, was also applied for each SES group separately. The results of these tests are comparable to the results of the complete matched sample. However, for most SES groups in contrast to the complete matched sample, no significant differences are observed between the treatment and control group for the number of shortage points and GPA for all courses.

Similar to the analysis without matching, we perform an OLS regression with clustered standard errors on the school level, including the DiD-estimator in the first model and control variables in the second model. The results in Table 7 indicate that also for the matched sample, there is a significant difference of .206 ($P<0.05$) standard deviation in average growth in favor of the participants. The analysis results also reveal a reduction of one quarter for the average effect. This reduction is not uncommon, given that a better control for selection is included by using matching in combination with DiD [50]. Less control for selection often leads to overestimating the effect [51].

When distinguishing between the three SES groups (Table 8), a more substantial reduction, about half the effect, is observed. The direction of the effect and the strength of the relationship for the different SES groups suggest some comparability with the DiD-analysis without matching. However, although the DiD-estimators indicate a stronger average growth in math GPA for the participants in all three SES groups, only the effect for the highest SES group is significant. While this analysis does not confirm the findings of the DiD-analysis for different SES groups without matching, the absence of this confirmation is likely due to a lack of power. Table 8 shows that the number of observations and schools is considerably reduced when the DiD-analysis with matching is performed separately for each SES group.

## Conclusion and discussion

Summer school programmes, specifically focussing on disadvantaged students for whom the summer setback is more pronounced, are standard practices in the United States. In contrast, the goal of the recently introduced summer school programmes in secondary education in the Netherlands is not to prevent summer setbacks but to remediate learning gaps, in subjects such as math, with the ultimate goal of avoiding grade retention. From a policy perspective, finding out if the effects of summer schools differ across socioeconomic backgrounds entails

**Table 8. OLS regression of the DiD-estimates for Z Math GPA separately for matched SES groups.**

| | SES-1 | | SES-2 | | SES-3 | |
|---|---|---|---|---|---|---|
| | Model 1 | Model 2 | Model 1 | Model 2 | Model 1 | Model 2 |
| Constant | 0.096 | 0.888 | 0.0520 | 1.583** | 0.0332 | 0.582 |
| | (0.070) | (0.601) | (0.058) | (0.774) | (0.071) | (0.598) |
| Summer school | -0.682*** | -0.703*** | -0.664*** | -0.745*** | -0.647*** | -0.697*** |
| | (0.094) | (0.094) | (0.096) | (0.093) | (0.078) | (0.093) |
| Time indicator | -0.017 | -0.017 | -0.018 | -0.018 | -0.027 | -0.027 |
| | (0.069) | (0.070) | (0.037) | (0.037) | (0.061) | (0.061) |
| Summer school * Time indicator | 0.127 | 0.127 | 0.144 | 0.144 | 0.216** | 0.216** |
| | (0.133) | (0.134) | (0.102) | (0.103) | (0.085) | (0.086) |
| Control variables | | yes | | yes | | yes |
| Observations | 1,636 | 1,636 | 1,522 | 1,522 | 1,358 | 1,358 |
| Number of clusters | 28 | 28 | 32 | 32 | 23 | 23 |
| R-squared | 0.045 | 0.1 | 0.038 | 0.125 | 0.031 | 0.194 |

* $p < 0.10$

** $p < 0.05$

*** $p < 0.01$. Standard errors in parentheses are clustered at the school level

Note: included control variables are: gender, age, grade level, education track, track advice, grade repetition, and year of participation

Note 2: Not all schools in the matched sample had participants in each SES group. Schools that do not have participants in a certain SES group are dropped from the analysis for that specific SES group, leading to a smaller number of schools (clusters) compared to the main DiD-analyses.

particular relevance, especially concerning worries about increasing inequalities of opportunities [1, 2] and shadow education [23]. Therefore, it is essential to carefully evaluate and design these programmes so that children with a lower socioeconomic background at least *also* benefit from participation in these programmes. Accordingly, this study aimed to investigate whether publicly funded summer school programmes not explicitly aimed at students from lower socioeconomic backgrounds could still be of substantive meaning for these students and, herewith, possibly play a role in decreasing inequalities of opportunities in education.

The descriptive statistics, as well as the results of the DiD and matching methods, all indicate a stronger average growth in math GPA for participants in the summer schools. The aforementioned observed effect is comparable to the modest effects observed in previous research [8, 16, 26, 32] This positive effect is observed for all three SES groups in the DiD-analysis without matching. For the lowest and middle SES group, the average growth in math GPA for participants is less strong than for participants in the highest SES group. These results contrast previous literature that observed more substantial effects for students with a lower socioeconomic background [33] and students with a middle-class background [32]. The robustness analysis using DiD on the matched sample confirms the positive effect for participants. However, when distinguishing between SES groups, the respective coefficients are not significant for the two lowest SES groups. Nonetheless, it must be noted that the matching procedure significantly reduces the number of observed schools and herewith students, especially when distinguishing between the different SES groups. Therefore, a lack of statistical power may also be at play and explain the absence of a significant difference. Overall, we can conclude that the results indicate a positive effect for summer school participants. However, the effect seems less strong for participants in the lowest two SES groups.

The literature regarding the summer learning gap suggests that children with a lower socioeconomic background (i.e., students in the lowest SES group) experience a stronger learning setback during the summer holiday period [11, 13, 14, 16]. From this perspective, the results of

this study indicate that while high-SES participants benefit more in terms of outcomes after the summer holiday, low- and middle-SES participants still benefit not only from the absence of the summer learning gap but also by showing stronger average growth. However, the question remains if the unintentional effect of high-SES students benefitting more from summer school participation does not still lead to additional inequalities.

Although the way we measured socioeconomic background has been used in previous research [41] it is not a very precise indicator, as it is not estimated at the individual level, which increases the chance of an ecological fallacy. To indicate whether the three groups' chosen threshold is sensitive to this and to check whether the results hold if less heterogeneous groups are created, different SES group categories were distinguished and tested as a robustness check. These results, were in accordance with the results presented in this paper, which gives confidence to our results, but of course, does not fully take away the need for caution when interpreting the results. Differences in socioeconomic background and herewith the access to different economic, social, and cultural resources are often measured with indicators for parents' education level, occupational status, or income, where parents' education level is found to be a stable predictor for education success [4]. Taking parents' education level as an indicator of socioeconomic background would make it possible to more precisely distinguish between different SES groups and the differential effects of summer school participation.

The possibility to more precisely distinguish between SES groups is also relevant regarding the differential effects observed between different SES groups. Although summer schools in the Netherlands are not specifically aimed at decreasing inequalities of opportunities in education, it may be possible that different SES groups have other needs that could easily be incorporated into the organization of the summer schools. While summer schools have the potential to have positive effects on all students and specifically students from lower socioeconomic backgrounds, there can be specific factors that make summer learning programmes more effective. Regarding duration, we know on average that the summer schools in the Netherlands had a shorter duration (average of seven days, maximum of two weeks) compared to the programmes in the United States (three to six weeks). This duration may explain why we found a weaker effect for students in the lowest two SES groups. For these students, the duration can be of more importance since their parents have fewer resources available to provide their children with summer learning activities. In the context of these components, a limitation of this study is that we have no information about the quality of the instruction in Dutch summer schools. It may be possible that the instruction quality was not aligned with the needs of different SES groups, especially since we know that primarily non-qualified teachers (that is, students still in teacher education) were teaching in the summer schools [38, 39]. These not (yet) qualified teachers may have less experience with adapting to the differential needs of different groups of students. Furthermore, given the institutional context of Dutch education, it is unlikely that students who participated in summer schools participated in additional activities throughout the remaining summer, considering that the decision to pass to the next grade was made immediately after the summer school. However, it could be possible that students, specifically high-SES students, who attended summer schools that were passed through to the next grade, did receive additional (private) tutoring at the start of the new school year to avoid further deficits. Looking further into quality differences between summer schools in relation to socioeconomic background and this possible confounding variable are important next steps when further investigating heterogeneous effects.

Although the analyses were carefully carried out using various analysis methods, some uncertainties regarding the results cannot entirely be ruled out. A major concern is the discussed selection bias. There may still be some unobserved characteristics that influence participation and performance in the subsequent school year, such as motivation or student-specific

SES (including parental education, for example). Another uncertainty concerns the mechanisms behind the effect on achievement after summer school participation. It may be possible that summer school participation not only influences learning but also affects non-cognitive skill development such as self-efficacy or signalling commitment, which influences achievement. Whether summer schools increase educational performance through learning or non-cognitive skill development is important in terms of the theoretical and practical implications of our findings. Lastly, the outcome measure used in this study is profoundly different from previous studies, making it harder to compare results. Although the GPA is comparable between students and not influenced by individual teachers, it is not a standardized test or a test designed as part of the experiment, as most previous studies used for their analyses. It is possible that the differences between our study and previous studies are due to the difference in the way the outcome is measured. Nonetheless, given the objectives of the summer schools themselves, this research contributed to the literature from the perspective of investigating if the effect of summer schools aimed at all students differs across different SES groups. From a general policy perspective, these findings are very relevant in the light of increasing worries about inequalities of opportunities in education, especially given the recent temporary school closures as a consequence of COVID-19. Because of this temporary school closure, the debate on summer schools is more vivid than ever in some countries (e.g. The Netherlands and Belgium), as schools strive to reduce the negative impact of the closure on students' learning development as much as possible.

The findings of this research point toward opportunities for future research taking into account a more precise indicator for SES, instruction quality, and possible differential needs of SES groups. Furthermore, the indicated differential effects for SES groups and the possibility of differential needs are essential to consider in policy decisions.

## Supporting information

**S1 Table. Description of the Dutch grading system.**
(PDF)

**S2 Table. OLS regression of the DiD estimate for Z Math GPA: SES group classification based on 1 standard deviation minus/plus mean.**
(PDF)

**S3 Table. OLS regression of the DiD estimate for Z Math GPA: SES group classification based on 0.5 standard deviation minus/plus mean.**
(PDF)

**S4 Table. OLS regression of the DiD estimate for Z Math GPA: SES group classification based on 4 equal groups.**
(PDF)

## Acknowledgments

Authors wish to thank Melline Somers and Tim Huijts for helpful comments on earlier versions of this paper and all involved schools for kindly providing data.

## Author Contributions

**Conceptualization:** Mélanie Monfrance, Carla Haelermans, Trudie Schils.

**Data curation:** Mélanie Monfrance, Carla Haelermans.

**Formal analysis:** Mélanie Monfrance, Carla Haelermans.

**Funding acquisition:** Carla Haelermans.

**Investigation:** Mélanie Monfrance, Carla Haelermans.

**Methodology:** Mélanie Monfrance, Carla Haelermans.

**Project administration:** Carla Haelermans.

**Supervision:** Carla Haelermans, Trudie Schils.

**Validation:** Carla Haelermans.

**Visualization:** Mélanie Monfrance.

**Writing – original draft:** Mélanie Monfrance.

**Writing – review & editing:** Mélanie Monfrance, Carla Haelermans, Trudie Schils.

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
