## [Decision Letter · Decision Letter 0]

21 Feb 2024

PONE-D-23-25229Effects of summer schools: who benefits the most?PLOS ONE

Dear Dr. Monfrance,

Thank you for submitting your manuscript to PLOS ONE. After careful consideration, we feel that it has merit but does not fully meet PLOS ONE’s publication criteria as it currently stands. Therefore, we invite you to submit a revised version of the manuscript that addresses the points raised during the review process. Please submit your revised manuscript by Apr 06 2024 11:59PM. If you will need more time than this to complete your revisions, please reply to this message or contact the journal office at plosone@plos.org. Please include the following items when submitting your revised manuscript:A rebuttal letter that responds to each point raised by the academic editor and reviewer(s). You should upload this letter as a separate file labeled 'Response to Reviewers'.A marked-up copy of your manuscript that highlights changes made to the original version. You should upload this as a separate file labeled 'Revised Manuscript with Track Changes'.An unmarked version of your revised paper without tracked changes. You should upload this as a separate file labeled 'Manuscript'.If applicable, we recommend that you deposit your laboratory protocols in protocols.io to enhance the reproducibility of your results. Protocols.io assigns your protocol its own identifier (DOI) so that it can be cited independently in the future. For instructions see: https://journals.plos.org/plosone/s/submission-guidelines#loc-laboratory-protocols. Additionally, PLOS ONE offers an option for publishing peer-reviewed Lab Protocol articles, which describe protocols hosted on protocols.io. Read more information on sharing protocols at https://plos.org/protocols?utm_medium=editorial-email&utm_source=authorletters&utm_campaign=protocols.

We look forward to receiving your revised manuscript.

Kind regards,

Shihe Fu, Ph.D.

Academic Editor

PLOS ONE

Journal Requirements:

Additional Editor Comments:I have read carefully your paper and I am very impressed by your careful and rigorous implementation of the DID method. The qualified reviewer had the same thought and suggested a test for a dynamic effect. If you have two periods of post-treatment test scores for students who participated in the 2015-16 summer schools, you can do a test to see if the treatment effect persists in the second year after the summer school. I also have a few minor suggestions:

1. One clarification question is how the summer school is funded. I understand it is funded publicly, but is it funded by local government or by the school? If it is funded by school, and if large schools are less financially constrained, then, the school finance capacity may confound the summer school effect. Similarly, another observable variable is student-teacher ratio.

2. Relatedly, since school characteristics may affect the treatment effect and many students enrolled in summer school in each school, it is feasible to include school fixed effects (school dummies) to control for unobserved, time-unvarying school characteristics, if you haven’t done so. I notice you clustered standard errors at the school level which is great.

3. Table 8: since splitting sample reduces sample size substantially, you can also use the triple difference: interacting summer school*time indicator with SES1 and SES3 dummy variables separately (the default is SES2 category). Another benefit of doing so is that you do not need to worry about the problem of a small number of clusters which could increase standard errors.

Given your paper is in great shape and the research design is very careful and convincing, I will not send out your revised version for referee and will accept the revised version directly.

Reviewers' comments:

Reviewer's Responses to Questions

**Comments to the Author**

1. Is the manuscript technically sound, and do the data support the conclusions?

Reviewer #1: Yes

2. Has the statistical analysis been performed appropriately and rigorously? 

Reviewer #1: Yes

3. Have the authors made all data underlying the findings in their manuscript fully available?

Reviewer #1: No

4. Is the manuscript presented in an intelligible fashion and written in standard English?

Reviewer #1: Yes

5. Review Comments to the Author

Reviewer #1: This paper examines the effect of summer school on math performance of participants, using data in Netherland. They exploit a DID estimation strategy, and find that summer school participation increases math scores for students with lowest and highest socioeconomic status.

The paper addresses an important issue with a sound research design. My main comment is that the authors should further adopt an event-study design to examine the dynamic effects of summer school participation. Does the positive effect persist, grow, or diminish over time? The current Figure 2 has already examined the pre-trend. It seems that the data enables an event-study estimation.

6. PLOS authors have the option to publish the peer review history of their article (what does this mean?). If published, this will include your full peer review and any attached files.

Reviewer #1: No

---

## [Author Response · Author response to Decision Letter 0]

21 Mar 2024

First, we would like to thank you for your thoughtful consideration of our manuscript. We appreciate your constructive comments and the opportunity to revise and address the points raised. We believe that we have been able to address all of your comments and hope that you will be satisfied by the respective revisions and replies. In the following, you find a point-by-point response to the points that you stressed in the decision email. 

Response to the main points mentioned by the editor

Thank you for reminding us about this. We have carefully reviewed and revised the manuscript to ensure that it aligns with PLOS ONE’s style requirements. 

We note that you have indicated that there are restrictions to data sharing for this study. PLOS only allows data to be available upon request if there are legal or ethical restrictions on sharing data publicly.

If there are ethical or legal restrictions on sharing a de-identified data set, please explain them in detail (e.g., data contain potentially identifying or sensitive patient information, data are owned by a third-party organization, etc.) and who has imposed them (e.g., a Research Ethics Committee or Institutional Review Board, etc.). Please also provide contact information for a data access committee, ethics committee, or other institutional body to which data requests may be sent.

There are indeed legal restrictions on data sharing for this study. This is because the data collection and management agreement with the schools does not legally allow us to make the data publicly available. The data have been archived and are accessible through our departments Data Management Team. We have updated the data statement in the revised submission and included contact information for access to the data. 

The updated data statement in the submission portal now reads as follows:

The data has been archived within the network of Maastricht University and is available through the Research Data Management (RDM) team that is part of our department (ROA, Maastricht University). The data management team ensures long-term data storage (at least 10 years) for all research projects within our department. In the manuscript, the following statement regarding the data availability can be shared:

“Data cannot be shared publicly because of legal reasons regarding privacy and confidentiality. Data are available from the Research Data Management team at ROA, Maastricht University (contact via rdm-roa@maastrichtuniversity.nl) for researchers who meet the criteria for access to confidential data.”

Please include your full ethics statement in the ‘Methods’ section of your manuscript file. In your statement, please include the full name of the IRB or ethics committee who approved or waived your study, as well as whether or not you obtained informed written or verbal consent. If consent was waived for your study, please include this information in your statement as well. 

We have now included an ethics statement in the data section of the manuscript and marked this as a change in the revised version of the manuscript. Please note that we have included the ethics statement in the ‘data section’ rather than the ‘methods section’ of the manuscript, as this is where we further discuss the data collection. Furthermore, we have updated the ethics statement in the submission portal. 

The ethics statement in the manuscript now reads as follows:

“Participants and parents were informed about the purpose of the data collection and were given the option to decline participation. For the data collection and data management, a Privacy Impact Assessment was conducted, and privacy measures were taken accordingly. The Graduate School of Business and Economics at Maastricht University granted an exemption from the Institutional Review Board (IRB) for the data collection.” 

Please include captions for your Supporting Information files at the end of your manuscript, and update any in-text citations to match accordingly.

We have now included captions for the supporting information files at the end of the manuscript and separately uploaded the supporting information files with matching file names in the revised submission. 

We have carefully examined the reference list and ensured its completeness. During this examination, we found and corrected an incorrectly cited reference, removing it from the list and marking the change in the manuscript. Furthermore, we have completed the references that refer to our own work that were anonymous in the previous version. Other than these adjustments, we only made the necessary changes to align with PLOS ONE’s referencing style requirements. 

I have read carefully your paper and I am very impressed by your careful and rigorous implementation of the DID method. The qualified reviewer had the same thought and suggested a test for a dynamic effect. If you have two periods of post-treatment test scores for students who participated in the 2015-16 summer schools, you can do a test to see if the treatment effect persists in the second year after the summer school.

Thank you for your thorough review and positive feedback on our paper. Regarding the suggestion for a dynamic effect test, although interesting, this is unfortunately not feasible in our case. We only received data at one point in time, which means we obtained historical data but not for later periods. Furthermore, due to the limitations outlined in the contract with the schools, obtaining additional data including additional post-treatment tests scores was not possible. 

One clarification question is how the summer school is funded. I understand it is funded publicly, but is it funded by local government or by the school? If it is funded by school, and if large schools are less financially constrained, then, the school finance capacity may confound the summer school effect. Similarly, another observable variable is student-teacher ratio.

Thank you for raising this very relevant matter. Upon revisiting the manuscript, we acknowledge that the funding process for the summer schools was not clearly explained. Summer schools were funded by the National government. The available funding, based on participating students, was equal for all schools. We included this information regarding the funding procedure in the section ‘Summer schools in the Netherlands’ in the manuscript and marked this as a change. The information regarding the funding procedure in the manuscript now reads as follows:

“Summer schools in the Netherlands were funded by the National government. Schools were eligible to apply for financial support in both the 2015-2016 and the 2016-2017 school year. The funding allocation was based on the number of participating students, with a fixed funding amount per participating student, where each school received support proportional to its budgeted student count. This funding model ensures that differences in financial capacity did not hinder schools from organizing summer school programmes.”

Relatedly, since school characteristics may affect the treatment effect and many students enrolled in summer school in each school, it is feasible to include school fixed effects (school dummies) to control for unobserved, time-unvarying school characteristics, if you haven’t done so. I notice you clustered standard errors at the school level which is great.

We appreciate your suggestion on this matter, and we have re-run the analyses, including school fixed effects. However, the results of these analyses are almost identical, aligning with our expectations given the nature of the DiD analysis which already controls for variables that remain constant over time. Therefore, we have chosen not to include these additional analyses in the paper. 

While re-running the analyses, we also noticed some small issues with the categorisation of the data and the clustering of the standard errors. We have therefore re-run all analyses and robustness checks to correct for these minor issues. This resulted in slight changes in the descriptive tables and results, particularly for the middle SES group. Nonetheless, the main results and overarching findings of the analyses remain the same. Following from this we have made minor adjustments in the manuscript text that have been marked as changes. 

Table 8: since splitting sample reduces sample size substantially, you can also use the triple difference: interacting summer school*time indicator with SES1 and SES3 dummy variables separately (the default is SES2 category). Another benefit of doing so is that you do not need to worry about the problem of a small number of clusters which could increase standard errors.

Thank you for your suggestion regarding Table 8. We acknowledge the suggestion to employ the triple interaction, and we have considered this approach in earlier versions of the manuscript. However, for reasons of readability and clarity of the analyses and results in the manuscript, we decided to split the sample according to SES group. In the main analysis, reducing the sample size does not seem to pose a significant issue for the results, and it eases interpretation. Furthermore, Table 8 refers to a robustness check, including matching. The sample size is mostly reduced here because not all schools in the matched sample had participants in each SES group, and the reduced sample size has little to do with splitting the sample (as shown by Table 6 in which we present the main analyses split by SES group). However, upon reflection, we realized that this aspect was not clearly explained in the table note. Consequently, we have made adjustments to the table note to improve clarity. The table note of Tables xx and 8 now reads as follows:

“Note 2: Not all schools in the matched sample had participants in each SES group. Schools that do not have participants in a certain SES group are dropped from the analysis for that specific SES group, leading to a smaller number of schools (clusters) compared to the main DiD-analyses.”

Thank you once more for your valuable comments and the opportunity to submit a revised manuscript. We hope that we have been able to deal with the comments successfully and are looking forward to hearing from you.

---

## [Editor Report · Decision Letter 1]

28 Mar 2024

Effects of summer schools: who benefits the most?

PONE-D-23-25229R1

Dear Dr. Monfrance,

We’re pleased to inform you that your manuscript has been judged scientifically suitable for publication and will be formally accepted for publication once it meets all outstanding technical requirements.

Kind regards,

Shihe Fu, Ph.D.

Academic Editor

PLOS ONE
---

## [Editor Report · Acceptance letter]

2 Apr 2024

PONE-D-23-25229R1 

PLOS ONE

Dear Dr. Monfrance, 

I'm pleased to inform you that your manuscript has been deemed suitable for publication in PLOS ONE. Congratulations! Your manuscript is now being handed over to our production team.

Kind regards, 

on behalf of

Dr. Shihe Fu 

Academic Editor

PLOS ONE